# Fire Acupuncture versus conventional acupuncture to treat spasticity after stroke: A systematic review and meta-analysis

Xuan Qiu[1,2]* , Yicheng Gao[1,2], Zhaoxu Zhang[3], Sijia Cheng[2], Shuangmei Zhang[4,5]

**1** Clinical Medical College of Acupuncture, Moxibustion and Rehabilitation, Guangzhou University of Chinese Medicine, GuangZhou, China, **2** Guangzhou University of Traditional Chinese Medicine, GuangZhou, China, **3** Department of Neurology, Peking University People's Hospital, Beijing, China, **4** Department of Rehabilitation, Cancer Hospital of the University of Chinese Academy of Sciences (Zhejiang Cancer Hospital), HangZhou, China, **5** Institute of Cancer and Basic Medicine (IBMC), Chinese Academy of Sciences, HangZhou, China

☺ These authors contributed equally to this work.
* qiuxuan20202020@163.com

**Data Availability Statement:** All relevant data are within the manuscript and its Supporting Information files.

## Abstract

### Background

Post-stroke spasm is currently a complex clinical problem that remains to be resolved. Due to its excellent efficacy and few side effects, clinicians have used fire acupuncture to treat post-stroke spasticity in China.

### Objectives

The purpose of this study was to evaluate the clinical efficacy of fire acupuncture compared with conventional acupuncture to treat post-stroke spasms and provide a detailed summary of the commonly used acupoints.

### Methods

Eight databases (MEDLINE/PubMed, Web of Science, the Cochrane database, EMBASE, CBM, CNKI, WanFang, and VIP) were searched for randomized controlled trials (RCTs) published from database inception through August 30, 2020. RCTs that compared fire acupuncture with conventional acupuncture as a treatment intervention for patients with spasticity after stroke were included. Revman 5.3 software was used to calculate risk ratios (RR) and standard mean differences (SMD) with 95% confidence intervals (CI). Methodological evaluation or critical appraisal of the included articles was assessed using RoB-2.

### Results

Sixteen studies with a total of 1,118 patients were included. Although according to the standards of the Rob 2.0 tool, most studies are considered to have some problems. Comprehensive analysis of the results revealed a consistent trend indicating several advantages of using fire needles compared to conventional acupuncture in treating post-stroke spasms, including the effective rate, recovery rate, and improvement of multiple scales represented

**Funding:** (1) Zhuweifeng full name:Zhuweifeng Guangdong Provincial Famous Chinese Medicine Studio Construction Project Grant numbers: Guangdong Chinese Medicine Office 2018(5) (2) Zhaoxu Zhang full name:2020 Major Project of Science and Technology Project of Guangdong Province Traditional Chinese Medicine Bureau Grant numbers:(20203015) the funder play an role in the preparation of the manuscript (3) Zhaoxu Zhang full name:Science and technology development plan of Shandong Province Grant numbers:(2017G006021) the funder play an role in the preparation of the manuscript.

**Competing interests:** The authors have declared that no competing interests exist.

by MAS. Concerning secondary outcomes, using the scales of FMA, BI, or NDS in this random model meta-analysis, fire acupuncture exhibited better performance compared to acupuncture [SMD = 2.27, 95%CI [1.40,3.13 (random-effects model) ], [SMD = 1.46,95% CI [1.03,1.90 (random-effects model)], and [SMD = 0.90, 95%CI [0.44,1.35 (random-effects model)], respectively, with moderately high heterogeneity. When the effective rate was used as an outcome in the subgroup analysis, fire needles performed better than conventional acupuncture with respect to damage to the upper or lower limbs, and the thickness and depth of acupuncture. When the modified Ashworth scale (MAS) was used as the outcome, and the damage occurred in the lower extremity, the acupuncture depth exceeded 15mm, or the duration of stroke was longer than six months, the fire needles did not perform better than conventional acupuncture, [SMD = 0.01, 95%CI [-0.47,0.48 (fix-effects model)], [SMD = 0.21 [-0.51,0.93(random-effects model)], and [SMD = 0.76, 95%CI [-0.08,1.60 (random-effects model)], respectively. The acupoints identified with the highest frequencies in this study were Yang-meridian, including LI11-Quchi (nine times), LI4-Hegu (seven times), and ST36-Zusanli (five times). Moreover, no serious adverse effects were reported in any of the studies included in this analysis.

## Conclusions

Despite several limitations, this was the first meta-analysis to focus on the treatment of post-stroke spasticity using fire needle acupuncture compared with conventional acupuncture. Our results confirmed that fire needles could provide a better clinical effect than conventional acupuncture, which will help standardize fire needle treatment strategies for post-stroke spasms.

## Introduction

Stroke is a major public health problem and ranked as the most common cause of disability [1]. Although recent medical advances have reduced stroke to the fourth cause of death worldwide [2], it still represents a condition that results in devastating physical disability, particularly due to the presence of spasticity [3]. Stroke is the primary cause of death in China [4, 5], and the most common post-stroke complication is spasticity [6]. The prevalence of post-stroke spasticity ranges from 30% to 80% in stroke survivors [7], with a 90% probability of occurrence approximately three weeks after a stroke event [8, 9].

Spasticity is a motor disorder associated with lesions of the central nervous system (CNS) that provoke different clinical syndromes, including spasms, clonus, or hypertonia. It is noteworthy that spasticity is associated with reduced functional independence and a fourfold increase in direct care costs during the first year after stroke [10]. Several therapeutic approaches have been proposed to manage spasticity, including central muscle relaxants (baclofen and baclosan) and peripheral muscle relaxants (xeomi) [11–14]. Although medications can relieve the spasms, the relief is not long-lasting, and severe side effects are associated with long-term use of these drugs, including cardiac arrhythmia, hyperkalemia, and amyostasia. These adverse side-effects must be taken into consideration, particularly with elderly patients. Therefore, current studies are focused on identifying alternative treatment strategies, including conventional acupuncture and fire acupuncture. These alternative treatment strategies are prevalent in China and have been incorporated into clinical practice.

Some evidence suggests that acupuncture (including electroacupuncture) could reduce spasticity associated with other CNS diseases [15]. The fire needle is an important component of acupuncture. Due to their excellent curative effect and reduced side effects, fire needles have been used recently by clinicians in China to treat post-stroke spasticity. The efficacy of acupuncture is widely recognized. It has been used to resolve functional recovery problems after CNS injury for many years in Asian countries and is increasingly popular in western countries [15]. However, the general knowledge of fire needles around the world is still insufficient. Additionally, the results of studies on the comparative effectiveness of fire needles and conventional acupuncture in treating spasticity in stroke survivors have been variable. Therefore, it was necessary to conduct a systematic review and meta-analysis of the existing literature to objectively evaluate the clinical efficacy and safety of fire acupuncture for spasms after stroke.

## Methods

This systematic review and meta-analysis was registered in the PROSPERO database at http://www.crd.york.ac.uk/PROSPERO(CRD42020188959) and followed the guidelines provided by the Preferred Reporting Items for Systematic Review and Meta-Analysis (PRISMA) statement [16].

The elaboration of the scientific question was based on the PICO strategy [17] considering:

P- Participants/population: Patients diagnosed with limb cramps after stroke will be focused on. No restrictions on gender, age, and ethnicity.

I- intervention(s):Intervention groups are treated with fire needle alone.

C- Comparator(s)/control: The control group used acupuncture instead of fire needle on the basis of any type of acupuncture type

O- Outcome(s): The main outcome indicators included the effective rate (ER), recovery rate (RR), and the modified Ashworth scale (MAS). Secondary outcomes: The secondary outcome indicators included one of the following, Fugl-Meyer (FMA), Barthel Index (BI), and Neurological Function Deficit Scale (NDS).

### Data sources and search strategies

Eight databases (MEDLINE/PubMed, Web of Science, the Cochrane database, EMBASE, CBM, CNKI, WanFang, and VIP) were searched for RCTs published from the database inception through August 2020. Various combinations of Medical Subject Headings (MeSH) and non-MeSH terms were used, including "fire needle," "red-hot needle," "heated needle," "needle," "acupuncture," "acupotomy," "stroke," "cerebrovascular accident," "spasm," "paraparesis," "spastic," and "spasticity after stroke," which were searched individually or in combination. Language, study population, or country restrictions were not applied. Moreover, we examined other relevant medical journals and magazines to identify literature not included in the electronic databases. The specific search strategy is provided in the S1 Appendix.

### Criteria for inclusion and exclusion

**Criteria for inclusion.**   Studies that met the following conditions were enrolled in the analysis. The studies covered patients diagnosed with stroke and did not take muscle relaxants or have increased muscle tone. The experimental group only used fire acupuncture as the intervention, while the control group was treated with conventional acupuncture or electroacupuncture. The type, thickness, and procedure used for fire acupuncture were not limited. The main outcome indicators included the effective rate, recovery rate, and scales used to assess the degree of spasticity of the extremities. The scales used to assess daily living activities and neurological deficits were used as the secondary outcome indicators. There was no

restriction on age, gender, course of the disease, and treatment location. The language of the published research was not limited.

**Criteria for exclusion.**   Studies were excluded based on the following conditions. Fire acupuncture was combined with other treatment methods, including traditional Chinese medicine, blood puncture, rehabilitation, Chinese herbal (patent) medicine, or other treatments in the intervention group. The control group, which was acupuncture or electroacupuncture, was combined with other complex therapies. The patients exhibited increased muscle tone caused by other diseases. Others exclusion criteria that were used included whether the study was retrospective, a review, or a case report, the patient baseline data were inconsistent the study used inappropriate random sequence generation methods, conference papers, or data were missing from the report with no reply from the corresponding author(s).

## Data collection, extraction, and management

A piloted data extraction form that has been discussed and developed by all the reviewers was assessed and extracted independently by two authors (QX and ZSM). A standardized form was used to extract data, including general information, study characteristic, participant characteristic, interventions characteristics, outcomes and so on. Any disagreement in data extraction was resolved by discussion or negotiation with a third arbitrator (ZWF). The data included in the study were extracted according to a pre-designed standardized table, including the first author, publication date, and treatment location. The patient information included age, course of the disease, gender, and sample size. The intervention information included selection and depth of acupuncture points, materials used for acupuncture, treatment frequency, adverse events, and whether follow-up examinations occurred. If the data are incomplete or other problems are encountered during data extraction, we contacted the author by phone or e-mail for additional information. Each eligible trial was assigned to a study ID in the following formats: the name of the first author + space + year of publication (e.g, Wang T 2019).

## Risk of bias assessment

The Cochrane Handbook for Systematic Reviews(RoB-2) was used to evaluate all studies in this analysis to determine the bias associated with each study. RoB 2 [18] -a revised Cochrane tool assessing risk of bias arising from five domains in randomised trials:(1) the randomisation process, (2)deviations from the intended interventions,(3) missing outcome data, (4) measurement of the outcome, (5) and selection of the reported result. Each domain a risk of bias (low risk, some concerns, or high risk) based on the domain algorithm, and made an overall judgment (low risk, some concerns or high risk) using the described criteria. According to RoB 2, risk-of-bias judgments for each domain have the following categories: low risk of bias, some concerns, or high risk of bias. Judgments are based on and summarise the answers to signalling questions. interior agreement will be assessed for each domain of bias and for the overall RoB judgement by juding Fleiss's Kappa scores [19, 20]. 44 45 We will group agreement as poor (0.00), slight (0.01–0.20), fair (0.21–0.40), moderate (0.41–0.60), substantial (0.61–0.80) or almost perfect (0.81–1.00) [21].

The STRICTA checklist was used to evaluate the quality of the research. The funnel chart was used to analyze potential publication bias.

## Data synthesis

All analyses were conducted with Review Manager V.5.3 software and Stata. If a meta-analysis is not possible, we provided a narrative summary of the results from individual studies. The relative risk (RR) was used to analyze dichotomous outcomes. The mean difference (MD) was

used to analyze continuous outcomes with the same unit. Otherwise, the standardized mean difference (SMD) was used. The uncertainty was expressed with 95% confidence intervals (95%CI). We measured heterogeneity using the I2 statistic. Fixed-effects model was used if heterogeneity is found. Random effect model was used where significant statistical heterogeneity exists. Heterogeneity was further explored using meta-regression with backward elimination to analyze the associations between treatment effect and the participant characteristics. Funnel plot was used to examine the potential for publication bias. We judged heterogeneity based on the p-value. When the I2 was less than or equal to 50%, we determined that the heterogeneity was within an acceptable range, and adopted a fixed-effect model for the meta-analysis. We concluded that the heterogeneity was high if the I2 is greater than 50% and used a random-effect model for data analysis. If the number of studies included in the analysis is sufficient, subgroup analysis was used to determine heterogeneity. If the number of included articles exceeds 10, we thought that meta regression can be used to find the source of the heterogeneity.

### Outcome measures

**Primary outcomes.**   The main outcome indicators included the effective rate (ER), recovery rate (RR), and the modified Ashworth scale (MAS). The ER and RR reflected the improvement before and after treatment. The MAS is considered to be an important tool to assess spasticity after stroke. The scale is divided into 0, 1, 1+, 2, 3, and 4, to achieve a total of six levels. A higher score indicated increased muscle spasticity (0 = none, 4 = most severe).

**Secondary outcomes.**   The secondary outcome indicators included one of the following, Fugl-Meyer (FMA), Barthel Index (BI), and Neurological Function Deficit Scale (NDS). The FMA covers complex content, including tendon reflexes, muscle coordination, finger grip, joint mobility, and others. The lower the score, the worse the condition (0 = none and 100 = most severe). The BI scale primarily reflected the activities of daily life (0 = not affected and 100 = most severe). The NDS mainly evaluated speech, consciousness, facial paralysis, limb function, and others (0 = none and 45 = most severe). The primary outcome indicators included the effective rate (ER), recovery rate (RR), and the modified Ashworth scale (MAS). The ER and RR reflected the improvement before and after treatment. The MAS scale is divided into 0, 1, 1+, 2, 3, and 4. The higher the score, the worse the degree of muscle spasticity (0 = none, 4 = most severe).

## Results

### Study selection

A total of 2,354 articles related to the topic were retrieved through a comprehensive database search, of which 1,097 articles were duplicates. Based on the premise of excluding all irrelevant data, a total of 17 RCTs were included in the analysis. One of the 17 articles was eliminated from the final set of studies because the data were incomplete, and the missing data could not be obtained from the study's corresponding author. Eventually, 16 articles, with a total of 1,118 patients, were included in the analysis [22–37]. The detailed screening process is shown in (Fig 1).

### Study characteristics

**Basic characteristics of the included studies.**   All trials were conducted in China and published in Chinese, and all studies were carried out based on traditional acupuncture theory. Among the included studies, four were master theses [25, 33, 34, 37]. All the control groups included in the studies received acupuncture treatment. Fifteen of the studies compared fire

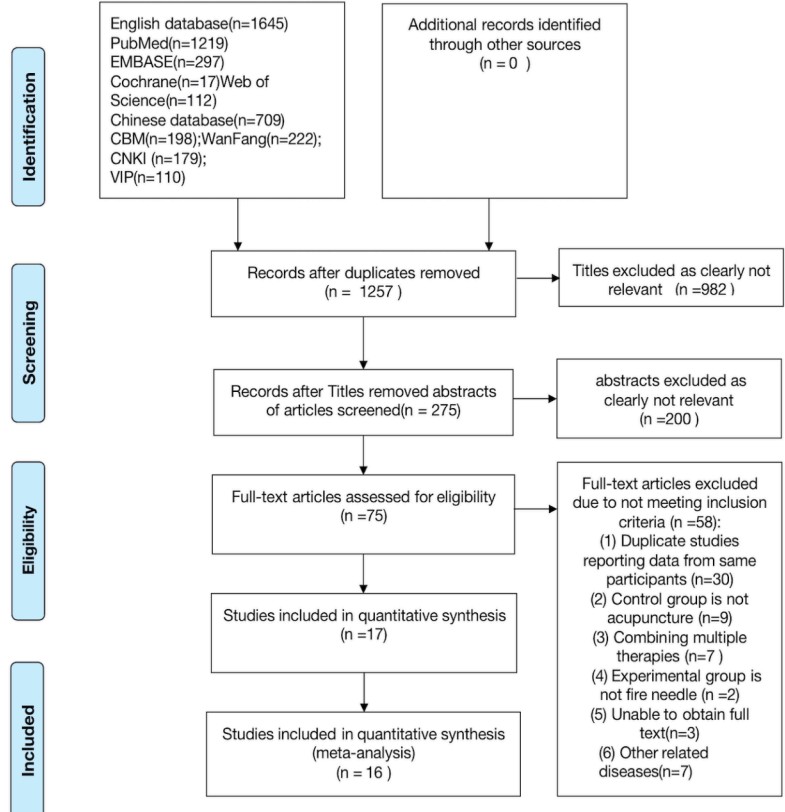

**Fig 1. Modified PRISMA flow diagram of included/excluded studies.**

needles with conventional acupuncture [21, 22, 25–27], and one study compared fire needle acupuncture with electroacupuncture [37]. The sixteen studies were published between 2005 and 2018. The ages of the patients ranged from 34 to 80 years. The course of the disease ranged from two months to one year. However, two studies did not report relevant information [23, 30]. Nine articles provided comprehensive information [21–24, 27, 31, 33, 34, 36, 37] regarding the type of stroke (intracerebral hemorrhage or cerebral infarction) exhibited by the patients. On the other hand, no specific information concerning the type of stroke was mentioned in four studies [25–27, 29, 30]. Three studies described the type of stroke that the patients experienced, but the report lacked detailed figures [28, 32, 35]. Eleven studies emphasized the state of the stroke, of which seven studies recorded that the patients were in the recovery period or sequelae after stroke [23, 25, 28, 31–33]. Three studies included patients who were in the recovery period, one study included participants that were in the sequelae period [36] and four studies did not report the related information [23, 25, 28, 31–34]. In seven studies, patients received basic medical treatment, including control of blood pressure, regulation of blood glucose, stabilization of blood lipids, and nutritional support [21–23, 25, 28, 31–32, 34].

In addition to receiving acupuncture, participants in four studies also participated in basic rehabilitation training [22, 24, 28, 30]. All studies described the specific sites of treatment. Five studies assessed the upper limbs [24, 25, 29, 31, 37], one study evaluated the fingers [22], and nine studies assessed the upper and lower limbs [19, 23–25, 27–29, 33]. Except for three studies that distinguished the severity of stroke [28, 36, 37], none of the other studies provided such a description. Fifteen trials [22–35, 37] used ER, and nine

studies [24, 25, 27, 28, 30, 32–35] used the MAS scale to assess the degree of improvement in spasticity after treatment. The FMA scale also was used in ten studies [23–28, 30, 32, 34, 35]. Six studies utilized the BI scale to assess the daily abilities of patients [23, 26, 28, 34, 35, 37]. No studies conducted follow-up assessments. None of the studies reported any fatal adverse events that resulted from acupuncture. The detailed characteristics of each study are shown in (Table 1).

**Details of the intervention groups in the included studies.** There were eleven studies [21–31] that provided the insertion depth of the fire needles, which ranged from 3mm to 30mm in ten studies. However, one study utilized shallow skin penetration to a depth of only 1mm to 3mm [26]. Concerning the types of fire needles used, they were approximately 0.35mm in diameter with lengths that ranged from 20 to 40 mm. The course of treatment ran between two weeks and one month. It was emphasized that a rest period was to be taken in the middle of the treatment. Additional details are shown in (Table 2). The acupuncture points used with high frequency in the sixteen studies were as follows (Table 3), LI11-Quchi (nine times), LI10-Shousanli (nine times), Extrapoint-baxie (seven times), SJ5-Waiguan (seven times), LI4-Hegu (seven times), SI3-Houxi (six times), ST-Zusanli (five times), LR3-Taichong (four times), SJ4-Yangchi (four times), and SP6-Sanyinjiao (four times). Among all the meridians, the large intestine meridian of the hand-yangming, the small intestine meridian of the taiyang, and the triple energizer meridian of the hand-shaoyang were the three most frequently used. These meridians are usually referred to as the three yang channels of the hand.

**STRICTA checklist for the included studies.** The STRICTA checklist is taken from the Standards for Reporting Interventions in Controlled Trials of Acupuncture [38]. Additional details are shown in Table 4. All studies [22–37] fully described the style of acupuncture that was used, the rationale for treatment, and the literature sources used to justify the rationale for acupuncture use. Concerning needling details, ten studies [25–34] recorded unilateral or bilateral usage of acupuncture points. Only one study [35] did not specify the number of acupuncture points. Five studies [22, 23, 35–37] did not describe the depth of acupuncture needle insertion. All eleven studies [25–27, 28–34, 37] discussed possible reactions that were caused by the use of acupuncture. Due to the particular forms of fire needles that were used, all studies explained the stimulation form of acupuncture. Only one study [22] did not record the needle retention time. Six studies did not introduce the type of acupuncture [22, 23, 29, 30, 35–36] two studies [29–37] did not describe the number of treatment sessions, and one trial [29] did not mention the treatment frequency. All eleven studies did not combine acupuncture with other therapies. Nine of the studies thoroughly described the setting and context for treatment, and two studies [29, 30] did not. None of the studies described the duration of relevant training for the acupuncture therapists. Also, only three studies [22, 29, 35] mentioned the training time and professional level for acupuncture therapists. All studies provided sources that justified the choice of control subjects.

**Study quality.** According to the criteria of the RoB 2.0 Tool, most of the studies are considered to have some concerns. The items that affect the quality of most studies are the blind implementation and missing outcome data, Only three presented low risk of bias in all the assessed domains [23, 29, 30], other studies have suggested some concerns to a certain degree in each items (Table 5), Fleiss's Kappa scores in our research Hinted substantia (Fleiss's Kappa scores = 0.75).

Funnel plot of publication bias. Using a funnel plot, the research team analysed publication bias in all included studies (Fig 2). The outcome suggested that there was little publication bias.

**Table 1. Detail of studies include.**

| First author (year) | Age range TG/CG (M ±SD) | Genger(M:F) TG/CG (M ±SD) | Sample size (TG/CG) | Duraton after Stroke (TG/CG) | Control intervention | Outcome measures | Intergroup differences | Follow-up |
|---|---|---|---|---|---|---|---|---|
| Peng A [22], 2017 | T:57.1±7.8 C:56.7±8.2 | T:15/11 C:14/12 | 26/26 | NR | AT:30 min,everyday,rest for 1 day after 6 consecutive treatment,30d | ER | P < 0.05 In favor of FA | NR |
| Yang [23], 2017 | T:60.5±5.8 C:61.8 ±6.4 | T:11/7 C:12/6 | 18/18 | 18.5±3.9/20.5± 4.3(w) | AT:10 min,every other day,1 session (1 session = 3 wk, one day rest between each weeks | ER;(2)FMA; (3) BI | (1)(2)(3)P < 0.05 In favor of FA | NR |
| Sang [24], 2017 | T:62.57±5.75 C:63.20±7.07 | T:19/11 C:17/13 | 30/30 | 168.70±56.99/ 166.17±87.02 (d) | EA:30 min,everyday,14d | (1)ER;(2);FMA(3)MAS | (1)(2)(3)P < 0.05 In favor of FA | NR |
| Liu [25], 2018 | NR | T:19/12 C:18/13 | 31/31 | 15-338/17-342 (d) | AT:30 min,every other day (1 session = 28d,a total of 14 treatment | (1)ER(2)FMA(3)MAS (wrist.elbow) | (1)(2)(3)P < 0.05 In favor of FA | NR |
| Chai [26], 2017 | T:61.90±3.18 C:58.93 ±11.93 | T:17/13 C:16/14 | 30/30 | 110.57 ±36.641/ 112.70±36.69 (d) | AT:30 min,everyday,4 weeks (6 times a week) | ER (2)FMA (3) BI | (1)(2)(3)P < 0.05 In favor of FA | NR |
| Wang [27], 2018 | T:64.87±8.18 C:65.67±7.52 | T:19/11 C17/13 | 30/30 | 99.83±32.31/ 99.83±32.31 (d) | AT:30 min,everyday,14d | (1)ER; (2)FMA; (3)MAS (wrist,elbow,keen,ankle); (4)NDS(median,ulnar nerve) | (1)(2)(3)(4) P < 0.05 In favor of FA | NR |
| Deng [28], 2017 | T:61. 4±6. 2 C:63. 2±7. 7 | T:25/19 C:22/20 | 44/42 | 5.9±2.3/6. 1± 2.6(m) | AT:30 min,every other day 2 session (1 session = 7 times) | (1)ER; (2)FMA; (3)BI;; (4) MAS; (5)NDS(median, ulnar nerve) | (1)(2)(3)(4) P < 0.05 In favor of FA | NR |
| Peng B [29], 2017 | T:66.5±1.2 C:67.6±2.3 | NC | 50/50 | NR | AT:30 min | ER | P < 0.05. In favor of FA | NR |
| Sheng [30], 2017 | T:55. 62 ± 4.19 C:56.21± 3.98 | T:16/15 C:17/14 | 31/31 | 185.62±63.19/ 185.48±64.22 (d) | AT:30min,everyday, (1session = 6 times,1 day rest between every session),30d | (1)ER;(2)FMA;(3)MAS (wrist, elbow, keen, ankle) | (1)(2)(3)P < 0.05 In favor of FA | NR |
| Xu [31], 2015 | T:58.3±7.8 C:57.4±8.1 | T:29/11 C:27/13 | 40/40 | 20.2±4.6/19.6 ±4.3(w) | AT:30 min,every other day 2 session (1 session = 2 weeks) | ER | P < 0.05. In favor of FA | NR |
| Wang [32], 2015 | T:52.3±21.4 C: 5 4.5±20.7 | T:22/18 C:20/20 | 40/40 | 175.6±94.6/ 18.72±88.6(d) | AT:30 min,every other day(Rest for 1 day after 6 consecutive treatment) | (1)ER; (2)FMA; (3)MAS | (1)(2)(3)P < 0.05 In favor of FA | 1 mo |
| Yuan [33], 2015 | T:64.31±5.62 C:66.25±4.12 | T:18/12 C:16/14 | 30/30 | 8.22±3.53 /7.71±4.20(m) | EA:20min,everyday,2 session (Rest 2 days after 5 consecutive treatment) | ER; (2) BI;(3)MAS; (4)NDS | (1)(2)(3)(4) P < 0.05.In favor of FA | NR |
| Liu [34], 2014 | T:63.97±9.66 C:67.47±9.32 | T:16/14 C:13/17 | 30/30 | 167.35±34.26/ 179.33+42.32 (d) | AT:30min,everyday,8 times (Rest for 1 day after 6 consecutive treatment) | ER; (2)BI; (3)FMA; (4)MAS | (1)(2)(3)(4)P 0.05 In favor of FA | NR |
| Zhao [35], 2013 | T:61.20±7.35 C:60.70±6.79 | T:23/17 C:21/19 | 40/40 | 2.76±1.05/3.02 ±1.41(m) | AT:30min,everyday,2 session (1 session = 9 times) | (1)ER; (2)FMA; (3)MAS; (4) NDS | (1)(2)(3)(4) P < 0.05 In favor of FA | NR |
| Chen [36], 2005 | NR | T:18/12 C:16/14 | 30/30 | 7.8/8(m) | AT:30 min,every other day 2m | BI | P < 0.05. In favor of FA | NR |
| Gao [37], 2004 | T:56±0.71 C:54.8±5.13 | T:23/7 C:20/10 | 30/30 | 250.13/245.6 (d) | AT: everyday | (1)ER; (2)NDS | (1)(2)P < 0.05 In favor of FA | NR |

NC = not record; EA = electroacupuncture; AT = acupuncture treatment; ER = effective rate; RR = Recovery rate; FMA = Fugl-Meyer; MAS = The modified Ashworth scale; CSI = Clinic Spastcity Index; BI = Barthel Index; NDS = neurological function deficit scale; TG = treatment group; CG = control group; M = mean; SD = standard deviation.

## Results of the meta-analysis

**Main outcomes for ER, RR, and MAS.** Twelve RCTs in this study used the MAS scale to calculate the effective rates. The meta-analysis revealed that when a fixed model was used, the

**Table 2. Details of experimental interventions.**

| First author (year) | points | Needle type | Depth of Insertion (TG/CG) | Treatment Frequency | Sessions (TG/CG) |
|---|---|---|---|---|---|
| Peng A [22], 2017 | SP10(*Xuehai*), SI3 (*Houxi*), GB44 (*Zuqiaoyin*), Extra-point (*Baxie*), LI10 (*Shousanli*), LI11 (*Quchi*), LI14 (*Bilao*), SJ5 (*Waiguan*), KI3 (*Taixi*), SJ4 (*Yangchi*) | NR | NR | Every two days | 30 days (Rest for 2 day after 5 comsec utive treatment) |
| Yang [23], 2017 | Wrist:SJ3 (*Zhongzhu*), SJ4 (*Yangchi*), LI4 (*Hegu*);Upper limb:LI10 (*Shousanli*), LI11 (*Quchi*), SJ5 (*Waiguan*);Lower limb:BL40 (*Weizhong*), BL39 (*Weiyang*), BL40(*Heyang*), BL57 (*Chengshan*), BL37(*Yinmen*), SP6(*Sanyinjiao*) | 0.35mm*(30~40) mm | NR | every other day | 1 session (1 session = 3 wk) |
| Sang [24], 2017 | LI10 (*Shousanli*), SJ10(*Tianjing*) | NR | 0.5–1 cm (5-10mm) | Every day | 14 d |
| Liu [25], 2018 | Extra-point (*Jiaji*)(C3-7、 T1-3) | 0.35mm*(20~40) mm | 0.5–1.5cun (17-33mm) | every ther day | 1 session 1 session = 28d,a total of 14 treatments) |
| Chai [26], 2017 | LI15(*Jianyu*), LI11 (*Quchi*), SJ5 (*Waiguan*), SI3 (*Yanglao*), GB34(*Yanglinquan*), SP6(*Sanyinjiao*), ST6(Zusanli),Ashi point | 0.5mm*(25~30) mm | 1-3mm | every day | 4 weeks (3 times a week) |
| Wang [27], 2018 | Upper limb:SJ5 (*Waiguan*), LI10 (*Shousanli*), LI15(*Jianyu*); Lower limb:ST6 (*Zusanli*), GB34(*Yanglinquan*),BL40 (*Weizhong*), LR3 (*Taichong*) | 0.4mm*40mm | 1.5cun (50mm) | every day | 14 d |
| Deng [28], 2017 | Upper limb:LI15(*Jianyu*), LI11 (*Quchi*), LI10 (*Shousanli*), SJ5 (*Waiguan*), LI4 (*Hegu*),HT1(*Jiquan*),LU5(*Chize*);Lower limb:GB30(*Huantiao*),ST6(*Zusanli*), GB34(*Yanglinquan*), LR3(*Taichong*),SP6(*Sanyinjiao*), KI6(Taixi),GB40(Qiuxu)L | 0.35mm*40mm | 15mm | every other day | 2 session (1 session = 7 times) |
| Peng B [29], 2017 | LI4 (*Hegu*),LI11 (*Quchi*),LI15(*Jianyu*) | NR | 3-15mm | NR | NR |
| Sheng [30], 2017 | LI4 (*Hegu*),LI11 (*Quchi*),LI15(*Jianyu*), LI10 (*Shousanli*), GB34(*Yanglinquan*), BL40 (*Weizhong*), LR3(*Taichong*), GB30 (*Huantiao*),HT1(*Jiquan*) | 0.45mm*40mm | 3-15mm | every two days | 30 d (Rest for 2 days after 5 consecut-ive treatment) |
| Xu [31], 2015 | DU20(*Baihui*),Extra-poin (*Taiyang*),GB20 (*Fengchi*), DU16 (*Fengfu*) HT1(*Jiquan*),LU5(*Chize*),PC3(*Quze*),PC6 (*Neiguan*),PC7(*Daling*),LI4(*Hegu*), SI3(*Houxi*),SP6 (*Sanyinjiao*),SP9(*Yinlingquan*),(LR3(*Taichong*),LR2(*Jimai*), KI10(*Yingu*),SP5(*Shangqiu*),GB40(*Qiuxu*) | NR | 10-20mm | every other day | 2 session (1 session = 2 weeks) |
| Wang [32], 2015 | LI15(*Jianyu*), LI14 (*Bilao*), LI11 (*Quchi*),LI10 (*Shousanli*), SJ5 (*Waiguan*), SJ(*Yangchi*),SI3(*Houxi*),LI4(*Hegu*),Extra-point (*Baxie*),SP9(*Yinlingquan*), SP6(*Sanyinjiao*),KI10 (*Yingu*),LR3(*Taichong*),GB44(*Zuqiaoyin*) | 0.40mm*45mm | 3-15mm | every two days | 30 d (Rest for 2 days after 5 consecut- tive treatment) |
| Yuan [33], 2015 | A:Upper limb:LI15(*Jianyu*), LI4(*Hegu*),HT3(*Shaohai*),LI3 (*Sanjian*), Lower limb:ST31(*Biguan*),ST36(*Zusanli*), LR3 (*Taichong*) B:Upperlimb:SI9(*Jianzhen*),LU5(*Chize*),LI10 (*Shousanli*),SI3(*Houxi*), Lower limb:GB30 (*Huantiao*), GB31 (*Fengshi*),BL57(*Chengshan*),BL60(*Kunlun*) Group A and Group B alternate | 0.30–0.35mm*20-75mm | 20-25mm for upper limb; 20-30mm for Lower limb | every day | 2session(1session = 2 weeks) (Rest for 2 days after 5 consecutive treatm ent) |
| Liu [34], 2014 | A:PC6(*Neiguan*), LU5(*Chize*), PC3(*Quze*), HT3(*Shaohai*), SI3(*Houxi*), LI10(*Shousanli*),LI11 (*Quchi*) B:PC2 (*Tianquan*),HT2(*Qingling*),LU3(*Tianfu*),LI15(*Jianyu*),LI14 (*Bilao*), SJ13(Naohui) Group A and Group B alternate | 0.65mm*50mm | 0.3cun (10mm) | every two days | 8 times |
| Zhao [35], 2013 | Extra-point (*Baxie*), Extra-point (*Shangbaxie*), ST36 (*Zusanli*),ST40(*Jiexi*), ST34(*Liangqiui*),ST32(*Futu*),Extra-point (*Bafeng*) | NR | NR | every three days | 2 session (1 session = 3 times) |
| Chen [36], 2005 | Upper limb:LI15(*Jianyu*),LI10(*Shousanli*),LI11 (*Quchi*),LI4 (*Hegu*),Extra-point (*Bafeng*);Lower limbs:ST36(*Zusanli*), ST34(*Liangqiui*),ST32(*Futu*),ST40(*Jiexi*), Extra-point (*Baxie*) | NR | NR | every other day | 2 m |
| Gao [37], 2004 | Jiaji(C4-7),LI15(*Jianyu*),SJ14(*Jianliao*), SI13(*Quyuan*),SI12 (*Bingfeng*), SI11(*Tianzong*),SJ10(*Tianjing*),LI5(*Yangxi*),SJ4 (*Yangchi*),SI3(*Houxi*), SJ9(*Sidu*), LI11 (*Quchi*),LI12 (*Zhouliao*),Extra-point (*Waibaxie*), | NR | NR | every other day | NR |

**Table 3. Frequency of main acupoints.**

| Upper limbs | Frequency | Lower limbs | Frequency |
|---|---|---|---|
| Quchi (LI11) | 9 | Zusanli (ST36) | 5 |
| Shousanli (LI10) | 9 | Taichong (LR3) | 4 |
| Baxie (Extra-point) | 7 | Sanyinjiao (SP6) | 4 |
| Waiguan (SJ5) | 7 | Weizhong (BL40) | 3 |
| Hegu (LI4) | 7 | Taixi (KI3) | 3 |
| Houxi (SI3) | 6 | Yinlinquan (SP9) | 3 |
| Yangchi (SJ4) | 4 | Yanglinquan (GB34) | 3 |
| Chize (LU5) | 4 | Zhaohai(KI6) | 2 |
| Bilao (LI14) | 3 | Qiuxu(GB40) | 2 |

fire needle group could significantly improve the post-stroke spasticity compared with the acupuncture group [RR = 1.51[1.36,1.66], P<0.001, Fig 3]. Seven studies with a total of 420 patients used the MAS scale to evaluate the recovery rate. The results of the fixed model showed that the therapeutic effect of fire needles was superior [RR = 2.59 [1.75, 3.84], P <0.001, Fig 4]. A total of 12 studies that included 720 patients used the MAS scale to assess changes before and after treatment for patients with spasticity after stroke. The random model results demonstrated that, compared with the acupuncture group, the fire needle group had a stronger correlation with the improved score [MD = 0.47, 95%CI [0.18, 0.77], P = 0.002, Fig 5]. All results are provided in (Table 6).

**Table 4. STRICTA, Standards for Reporting Interventions in Controlled Trials of Acupuncture.**

| Study | Acupuncture rationale | | | Needling details | | | | | | | Treatment regime | | Cointerventions | | Practitioner background | | | Control intervention | | |
|---|---|---|---|---|---|---|---|---|---|---|---|---|---|---|---|---|---|---|---|---|
| First author(year) | 1a | 1b | 1c | 2a | 2b | 2c | 2d | 2e | 2f | 2g | 3a | 3b | 4a | 4b | 5a | 5b | 5c | 6a | 6b | 6c |
| Peng A [22], 2017 | YES | YES | YES | NO | YES | NO | NO | YES | NO | NO | YES | YES | NO | YES | NO | YES | YES | YES | NO | YES |
| Yang [23], 2017 | YES | YES | YES | NO | YES | NO | NO | YES | YES | YES | YES | YES | NO | YES | NO | NO | YES | NO | YES | YES |
| Sang [24], 2017 | YES | YES | YES | NO | YES | YES | NO | YES | YES | NO | YES | YES | NO | YES | NO | NO | YES | NO | YES | YES |
| Liu [25], 2018 | YES | YES | YES | YES | YES | YES | YES | YES | YES | YES | YES | YES | NO | YES | NO | NO | YES | NO | YES | YES |
| Chai [26], 2017 | YES | YES | YES | YES | YES | YES | YES | YES | YES | YES | YES | YES | NO | YES | NO | NO | YES | NO | YES | YES |
| Wang [27], 2018 | YES | YES | YES | YES | YES | YES | NO | YES | YES | YES | YES | YES | NO | YES | NO | NO | YES | NO | YES | YES |
| Deng [28], 2017 | YES | YES | YES | YES | YES | YES | YES | YES | YES | YES | YES | YES | NO | YES | NO | NO | YES | NO | YES | YES |
| Peng B [29], 2017 | YES | YES | YES | YES | YES | YES | YES | YES | YES | NO | NO | NO | NO | NO | NO | YES | YES | NO | NO | YES |
| Sheng [30], 2017 | YES | YES | YES | YES | YES | YES | YES | YES | YES | YES | YES | YES | NO | NO | NO | NO | YES | NO | YES | YES |
| Xu [31], 2015 | YES | YES | YES | YES | YES | YES | YES | YES | YES | NO | YES | YES | NO | YES | NO | NO | YES | NO | YES | YES |
| Wang [32], 2015 | YES | YES | YES | YES | YES | YES | YES | YES | YES | YES | YES | YES | NO | YES | NO | YES | YES | NO | YES | YES |
| Yuan [33], 2015 | YES | YES | YES | YES | YES | YES | YES | YES | YES | YES | YES | YES | NO | YES | NO | NO | YES | NO | YES | YES |
| Liu [34], 2014 | YES | YES | YES | YES | YES | YES | YES | YES | YES | YES | YES | YES | NO | YES | NO | NO | YES | NO | YES | YES |
| Zhao [35], 2013 | YES | YES | YES | NO | NO | NO | NO | YES | YES | NO | YES | YES | NO | YES | NO | NO | YES | NO | NO | YES |
| Chen [36], 2005 | YES | YES | YES | NO | YES | NO | YES | NO | YES | NO | YES | YES | NO | YES | NO | NO | YES | NO | NO | YES |
| Gao [37], 2004 | YES | YES | YES | NO | YES | NO | NO | YES | YES | YES | NO | YES | NO | YES | NO | NO | YES | NO | NO | YES |

STRICTA, Standards for Reporting Interventions in Controlled Trials of Acupuncture;1a, style of acupuncture; 1b, rationale for treatment (eg, syndrome patterns, segmental levels, trigger points) and individualisation if used; 1c, literature sources to justify rationale; 2a, points used (unilateral/bilateral); 2b, numbers of needles inserted; 2c, depths of insertion (eg, cun or tissue level); 2d, responses elicited (eg, de qi or twitch response); 2e, needle stimulation (eg, manual or electrical); 2f, needle retention time; 2g, needle type (gauge, length, and manufacturer or material); 3a, number of treatment sessions; 3b, frequency of treatment; 4a, other interventions (eg, moxibustion, cupping, herbs, exercises, lifestyle advice); 4b, setting and context of treatment, including instructions to practitioners, and information and explanations to patients; 5a, duration of relevant training; 5b, length of clinical experience; 5c, expertise in specific condition; 6a, intended effect of control intervention and its appropriateness to research question and, if appropriate, blinding of participants (eg, active comparison, minimally active penetrating or non-penetrating sham, inert); 6b, explanations given to patients of treatment and control interventions, details of control intervention (precise description, as for item 2 above, and other items if different); 6c, sources that justify choice of control; No, no details report; Yes, details reported.

**Table 5. Assessment of risk of bias for all included studies using the revised of bias tool (Rob 2.0).**

| Studies | Randomization | Intervention | Missing Data | Outcome measurement | Reported results | Overall Risk |
|---|---|---|---|---|---|---|
| Peng A [22], 2017 | High | Some concerns | Some concerns | Some concerns | Low | High |
| Yang [23], 2017 | Low | Low | Low | Low | Low | Some concerns |
| Sang [24], 2017 | High | Low | Some concerns | Some concerns | Low | High |
| Liu [25], 2018 | Some concerns | Low | Low | Low | Low | Some concerns |
| Chai [26], 2017 | Low | Low | Low | Low | Low | Low |
| Wang [27], 2018 | Low | Low | Low | Low | Low | Low |
| Deng [28], 2017 | Some concerns | Low | Low | Low | Low | Some concerns |
| Peng B [29], 2017 | Some concerns | Some concerns | Low | Some concerns | Low | High |
| Sheng [30], 2017 | Low | Low | Some concerns | Low | Low | Some concerns |
| Xu [31], 2015 | Some concerns | Some concerns | Some concerns | Low | Low | Some concerns |
| Wang [32], 2015 | Low | Low | Some concerns | Low | Low | Low |
| Yuan [33], 2015 | Low | Low | Low | Low | Low | Low |
| Liu [34], 2014 | Low | Low | Low | Low | Low | Some concerns |
| Zhao [35], 2013 | Some concerns | Low | Some concerns | Low | Low | Some concerns |
| Chen [36], 2005 | Some concerns | Low | Some concerns | Low | Low | Some concerns |
| Gao [37], 2004 | Some concerns | Low | Some concerns | Low | Low | Some concerns |

Low: Low risk of bias

High: High risk of bias

Some concerns: Some concerns of risk of bias

## Subgroup analysis

Based on the subgroup analysis of stroke-injured limbs on the ER side, the random model results showed that fire needle therapy for the upper and lower limbs produced significant

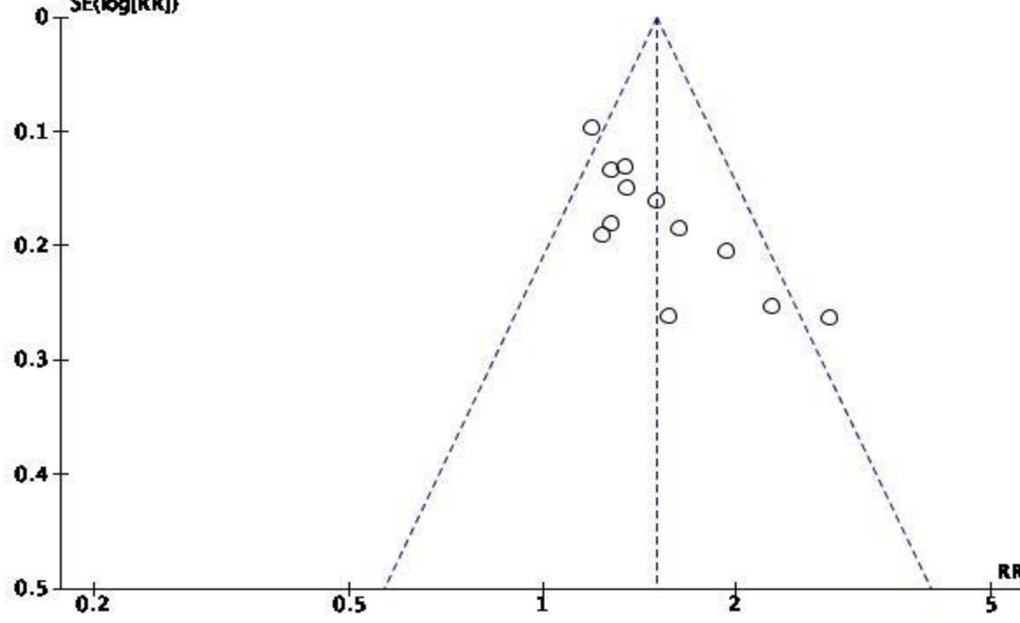

**Fig 2. Funnel plot of studies comparing fire acupuncture after stroke in ER.**

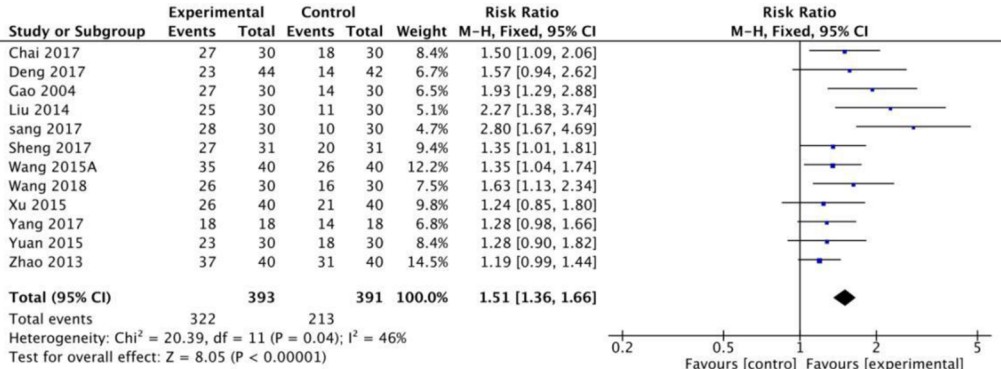

**Fig 3. Meta-analysis of fire acupuncture versus acupuncture for spasticity after stroke in ER.**

improvements, [RR = 1.71 [1.27, 2.30] and RR = 1.37 [1.11, 1.70], Fig 6], respectively. When using MAS to evaluate the degree of spasm in limbs injured by stroke, the fixed model results revealed that fire needles produced better results than conventional acupuncture in reducing the MAS score for the upper limbs [SMD = 0.50, 95%CI [0.28, 0.72], Fig 7]. However, the lower limbs did not show significant improvement [SMD = 0.01, 95% CI [-0.47, 0.48], Fig 7].

We also conducted a subgroup analysis based on the thickness of the fire needles and acupuncture depth. From the perspective of efficiency, fire needles were better than acupuncture to some extent with low heterogeneity, regardless of the depth or thickness changes in the subgroup analysis. When MAS was used to evaluate the improvement based on acupuncture depth, the random model results demonstrated that the fire needles were significantly deeper compared to conventional acupuncture, where the acupuncture depth was 3 to 15mm [SMD = 0.54, 95% CI [0.12, 0.95], Fig 8]. It is worth noting that when the acupuncture depth exceeded 15mm, the fire needles were not superior to acupuncture [SMD = 0.21, 95% CI [-0.51, 0.93], Fig 8].

We also extracted additional information from the studies included in the meta-analysis. When combined with the stroke duration and the extent of improvement as assessed using MAS, the subgroup analysis established that the extent of improvement for the fire needle scale was superior to acupuncture [SMD = 0.38, 95% CI [0.05, 0.70], Fig 9] when the stroke duration was less than six months. As expected, there was no difference between fire needles and conventional acupuncture when the duration of stroke was longer than six months [SMD = 1.14, 95% CI [-0.49, 2.78], Fig 9]. All results are presented in (Table 6).

**Secondary outcomes for FMA, BI, and NDS.** A random model meta-analysis was used to assess the secondary outcomes. Using any of the scales, FMA, BI, or NDS, revealed that fire

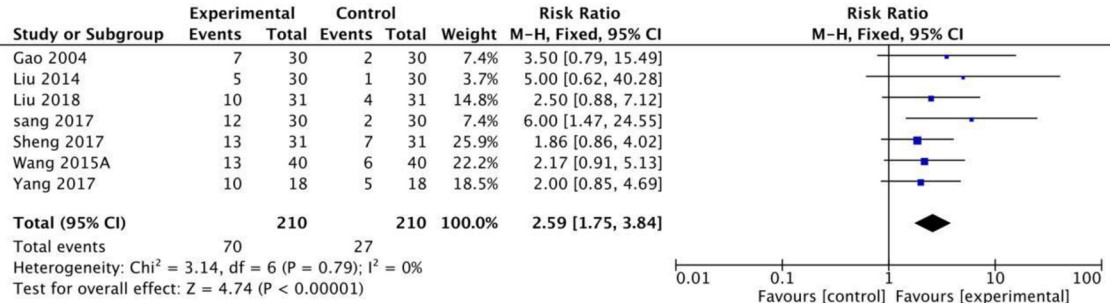

**Fig 4. Meta-analysis of fire acupunture versus acupuncture for spasticity after stroke in RR.**

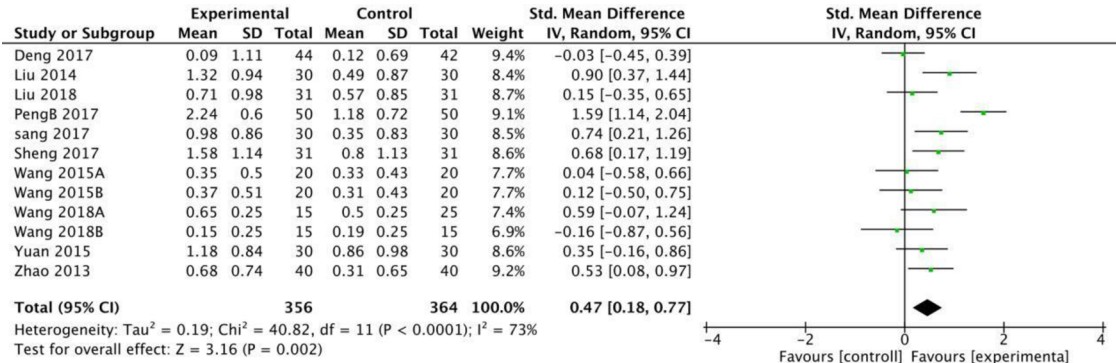

**Fig 5. Meta-analysis of fire acupuncture versus acupuncture for spasticity after in MAS.**

acupuncture exhibited better performances compared to conventional acupuncture [SMD = 2.27, 95% CI [1.40, 3.13], Fig 10], [SMD = 1.46, 95% CI [1.03, 1.90], Fig 11], and [SMD = 0.90, 95% CI [0.44, 1.35], Fig 12], respectively, with moderately high heterogeneity. All results are presented in (Table 6).

## Results of meta-regression

We initially judged that the source of heterogeneity may come from two major aspects. One is the difference related to stroke information, such as the type of stroke (intracerebral hemorrhage or cerebral infarction or both), and the influence of the location of spasm after stroke (Whole body, upper extremity or lower extremity), The second is the difference from the specific implementation process of fire acupuncture, such as the number of acupuncture points, the depth of acupuncture, and the frequency of treatment. All of the above differences may be the source of heterogeneity. Therefore, we performed meta-regression on the following factors. The results suggested that none of the above factors are the cause of heterogeneity (P>0.05), the results showed in (Table 7).

## Discussion

Joint convulsion, deformities, and muscle atrophy caused by stroke always result in clinical symptoms that include motor dysfunction, joint swelling, pain, and numbness. These symptoms reduce a patient's quality of life and produce severe physical, psychological, and economic burdens on patients, leading to depression, low self-esteem, despair, and suicidal thoughts [39, 40]. More importantly, stroke survivors may have reduced motivation to pursue rehabilitation training due to spasticity, which, in turn, has negative effects on their recovery outcomes [41, 42]. Multiple studies have recognized the efficacy of acupuncture for stroke sequelae, including relief of anxiety, and improved quality of life, especially for stroke patients [43]. Several meta-analyses focused on clinical practice have demonstrated that acupuncture exerts a beneficial effect in neurological and motor function recovery, including increased balance and muscle strength, and decreased spasticity [44, 45]. Clinical experience has indicated that fire-needle treatment takes less time, requires fewer visits, has more rapid results, and fewer side effects compared to chemical medicinal alternatives [46] At the same time, related clinical research has reported that fire-needle therapy was effective for sequelae of apoplexy [47, 48]. Most of the literature that was included in this study was published after 2015. Thus, the clinical and scientific focus on the use of fire needles has increased in China recently, confirming that fire needles are a reliable and reproducible treatment for post-stroke spasms.

**Table 6. Meta-analysis of the effects of fire acupuncture vs. electroacupuncture or acupuncture.**

| Outcomes or Subgroup 1.1ER | Studies 12 | Participants 784 | Statistical Method Risk Ratio (M-H, Fixed, 95% CI) | Effect Estimate 1.51 [1.36,1.66] | P P<0.00001 | Heterogeneity P=0.04;I²=46% |
|---|---|---|---|---|---|---|
| 1.2ER(for limbs) | 8 | 434 | Risk Ratio (M-H, Random, 95% CI) | 1.60 [1.29,1.97] | P<0.00001 | P=0.004;I²=66% |
| 1.2.1ER(for upper limbs) | 6 | 352 | Risk Ratio (M-H, Random, 95% CI) | 1.71 [1.27,2.30] | P=0.0005 | P=0.001;I²=75% |
| 1.2.2ER(for lower limbs) | 2 | 82 | Risk Ratio (M-H, Random, 95% CI) | 1.37 [1.11,1.70] | P=0.004 | P=0.50;I²=0% |
| 1.3ER(for the thickness of FA) | 10 | 566 | Risk Ratio (M-H, Fixed, 95% CI) | 1.46 [1.30, 1.64] | P<0.00001 | P=0.75;I²=0% |
| 1.3.1ER(≤0.35mm*(20~40) mm) | 4 | 244 | Risk Ratio (M-H, Fixed, 95% CI) | 1.42 [1.19, 1.70] | P<0.0001 | P=0.79;I²=0% |
| 1.3.1ER(>0.35mm*(20~40) mm) | 6 | 322 | Risk Ratio (M-H, Fixed, 95% CI) | 1.49 [1.28, 1.73] | P<0.00001 | P=0.46;I²=0% |
| 1.4ER (for the depth of FA) | 11 | 662 | Risk Ratio (M-H, Fixed, 95% CI) | 1.53 [1.37, 1.71] | P<0.00001 | P=0.23;I²=22% |
| 1.4.1ER(<3mm) | 1 | 60 | Risk Ratio (M-H, Fixed, 95% CI) | 1.50 [1.09, 2.06] | P=0.01 | Not applicable |
| 1.4.2ER(3-15mm) | 7 | 462 | Risk Ratio (M-H, Fixed, 95% CI) | 1.57 [1.38, 1.80] | P<0.00001 | P=0.05;I²=52% |
| 1.4.3ER(>15mm) | 3 | 140 | Risk Ratio (M-H, Fixed, 95% CI) | 1.39 [1.09, 1,79] | P=0.009 | P=0.61;I²=0% |
| 1.5RR(for the whole body) | 7 | 420 | Risk Ratio (M-H, Fixed, 95% CI) | 2.59 [1.75, 3.84] | P<0.00001 | P=0.79;I²=0% |
| 1.6MAS(for the whole body) | 12 | 720 | Std.Mean Difference (IV, Random, 95% CI) | 0.47 [0.18, 0.77] | P=0.002 | P<0.0001;I²=73% |
| 1.7MAS | 8 | 401 | Std. Mean Difference (IV, Fixed, 95% CI) | 0.41 [0.21, 0.61] | P<0.0001 | P=0.15;I²=35% |
| 1.7.1MAS(for upper limbs) | 6 | 332 | Std. Mean Difference (IV, Fixed, 95% CI) | 0.50 [0.28, 0.72] | P<0.00001 | P=0.22;I²=29% |
| 1.7.2MAS(for lower limbs) | 2 | 70 | Std. Mean Difference (IV, Fixed, 95% CI) | 0.01 [-0.47, 0.48] | P=0.98 | P=0.57;I²=0% |
| 1.8MAS(the course of disease) | 10 | 500 | Std.Mean Difference (IV, Random, 95% CI) | 0.50 [0.15, 0.84] | P=0.004 | P=0.0003;I²=71% |
| 1.8.1MAS(≤6m) | 8 | 408 | Std.Mean Difference (IV, Random, 95% CI) | 0.38 [0.05, 0.70] | P=0.02 | P=0.01;I²=61% |
| 1.8.2MAS(>6m) | 2 | 92 | Std.Mean Difference (IV, Random, 95% CI) | 1.14 [-0.49, 2.78] | P=0.17 | P=0.001;I²=71% |
| 1.9MAS(the depth of FA) | 10 | 570 | Std.Mean Difference (IV, Random, 95% CI) | 0.48 [0.11, 0.85 | P=0.01 | P<0.00001;I²=78% |
| 1.9.1MAS(3-15mm) | 8 | 510 | Std.Mean Difference (IV, Random, 95% CI) | 0.54 [0.12, 0.95] | P=0.01 | P<0.00001;I²=81% |
| 1.9.2MAS(>15mm) | 2 | 60 | Std.Mean Difference (IV, Random, 95% CI) | 0.21 [-0.51, 0.93] | P=0.57 | P=0.16;I²=50% |
| 1.10FMA | 7 | 418 | Std.Mean Difference (IV, Random, 95% CI) | 2.27 [1.40, 3.13] | P<0.00001 | P<0.00001;I²=92% |
| 1.11BI | 4 | 216 | Std.Mean Difference (IV, Random, 95% CI) | 1.46 [1.03, 1.90] | P<0.00001 | P=0.11;I²=51% |
| 1.12NDS | 3 | 180 | Std.Mean Difference (IV, Random, 95% CI) | 0.90 [0.44, 1.35] | P=0.0001 | P=0.11;I²=54% |

ER = effective rate; RR = Recovery rate; FMA = Fugl-Meyer; MAS = The modified Ashworth scale; CSI = Clinic Spastcity Index;

BI = Barthel Index; NDS = neurological function deficit scale.

Therefore, we conducted this systematic review and meta-analysis of RCTs to summarize the safety and efficacy of fire-needle therapy versus conventional acupuncture used to treat post-stroke spasticity with respect to recovery outcomes.

Comprehensive analysis of our results presented a consistent trend that the use of fire needles was advantageous compared to conventional acupuncture in treating post-stroke spasms. The benefits included improvements in the effective rate, recovery rate, and improvements based on multiple scales represented by MAS. Moreover, there were no reports of serious adverse effects in any of the included studies, such as fainting, dizziness, or unstable blood

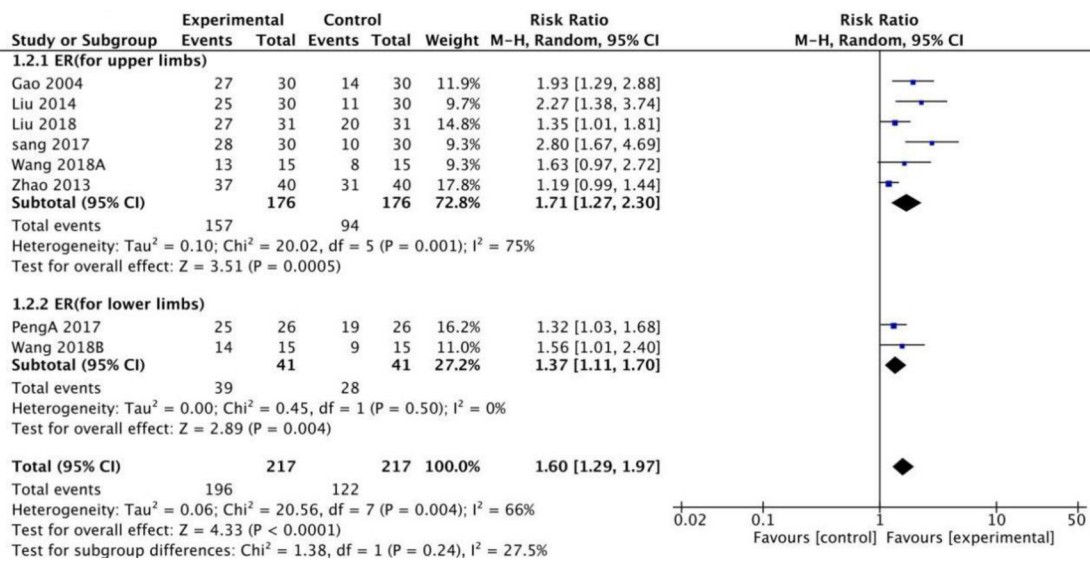

**Fig 6. Meta-analysis of fire acupuncture versus acupuncture for spasticity after stoke.**

pressure. This was similar to many clinical studies that have demonstrated the efficacy of conventional acupuncture on multiple sequelae of stroke [43]. In fact, guidelines for adult stroke rehabilitation and recovery recommend acupuncture for the treatment of stroke spastic paralysis. Therefore, we used conventional acupuncture as a control to comprehensively compare the advantages and disadvantages of fire acupuncture and provide valuable clinical evidence to show that fire acupuncture has unique advantages in relieving stroke spasms. Our results confirmed that fire needle treatment for post-stroke spasticity exerted better clinical effects compared to conventional acupuncture, which was consistent with previous clinical experiences and many current research conclusions [46] Also, the degree of improvement in the scores from FMA, BI, and NDS reflected that fire acupuncture decreased spasticity, and improved balance, accuracy, range of motion, and maintained stability in patients who had experienced a stroke.

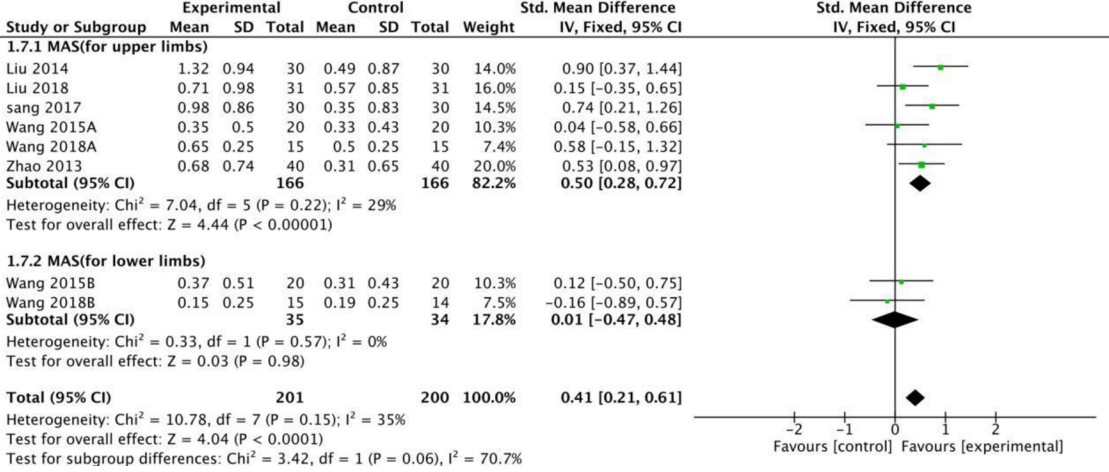

**Fig 7. Meta-analysis of fire acupuncture versus acupuncture for spasticity after stoke according to region.**

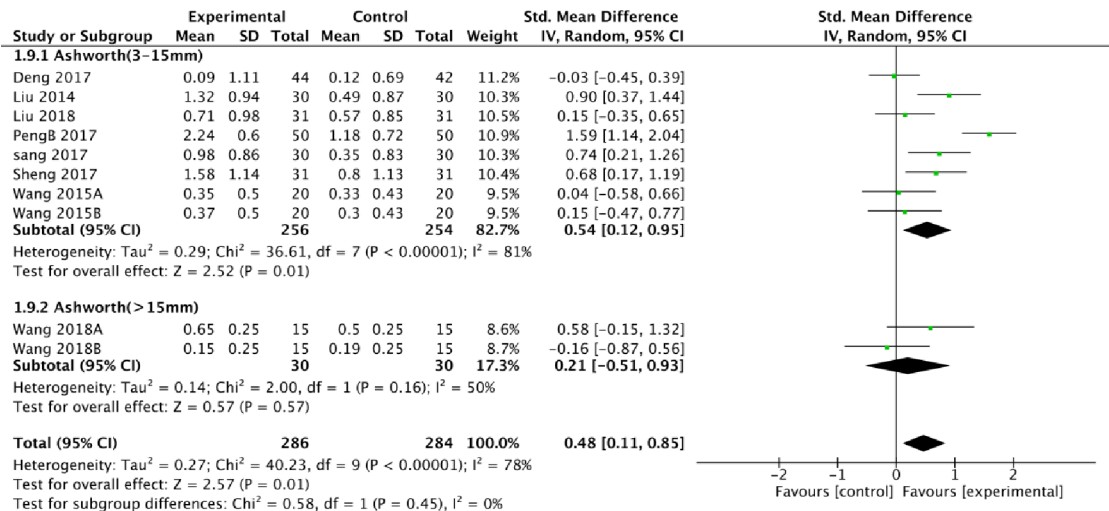

**Fig 8. Meta-analysis of fire acupuncture versus acupuncture for spasticity after stoke according to depth of acupuncture in MAS.**

One important finding was the high degree of heterogeneity among the different types of fire needles that were examined. There was no significant difference in the clinical efficacy of fire needles compared with conventional acupuncture with respect to the thickness of the fire needle diameter. Subgroup analysis revealed that when the fire needle depth exceeded 15mm, the fire needle was more efficient compared to conventional acupuncture [RR = 1.39, 95%CI ([1.09, 1,79], P = 0.009]. However, the improvement in the MAS score is not significant. [SMD = 0.21, 95% CI (-0.51, 0.93), P = 0.57].

The high heterogeneity of some results in the article cannot be ignored. Therefore, we conducted a meta-regression to find the source of heterogeneity. Based on clinical experience, we analyzed the following factors that may cause high heterogeneity, such as the type of stroke, and the location of spasm after stroke, the number of acupuncture points, the depth of acupuncture, and the frequency of treatment. However, the results of meta-regression suggest that

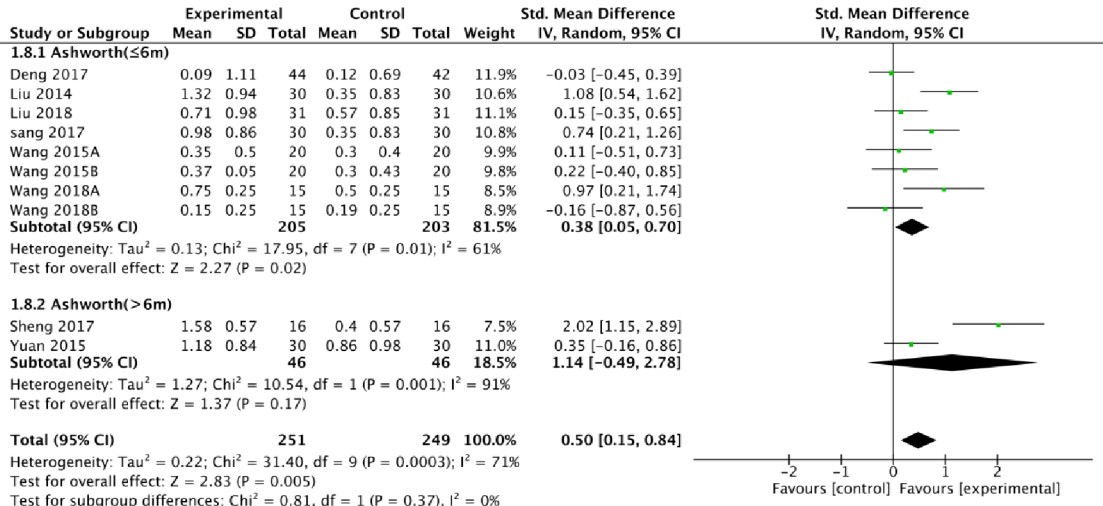

**Fig 9. Meta-analysis of fire acupuncture versus acupuncture for spasticity after stoke according to the course of disease.**

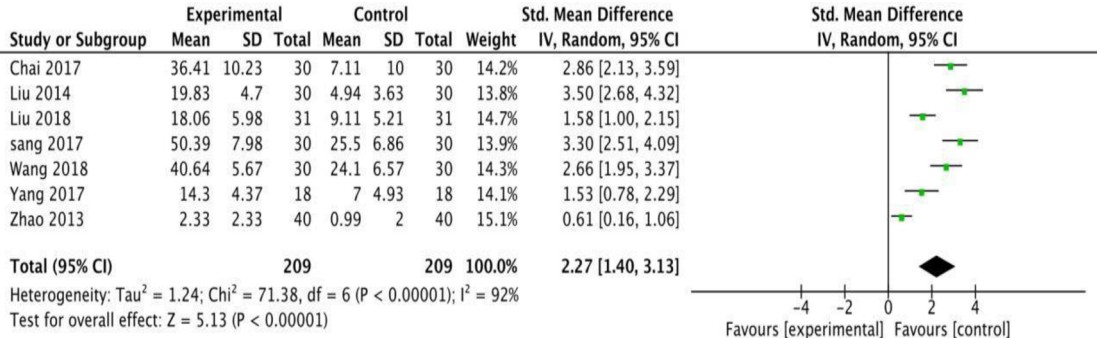

**Fig 10. Meta-analysis of fire acupuncture versus acupuncture for spasticity after stroke in FMA.**

none of the above factors is the cause of heterogeneity, but we can not rule out the influence of other uncontrollable factors, such as the period of stroke and the severity of stroke. Unfortunately, due to the fact that there are many missing values for these two factors in the included article, meta-regression cannot be performed, Therefore, we cannot analyze the impact of these two factors on heterogeneity. Future research should be as complete as possible to standardize the clinical application of fire needles and provide evidence for its further promotion.

Stroke is an acute cerebrovascular event, but the sequelae are chronic and persistent. Therefore, we conducted a subgroup analysis of studies that provided data on the course of stroke using MAS as a measurement indicator. Not surprisingly, fire needles exhibited excellent effects when the stroke duration was less than six months. However, when the stroke duration was longer than six months, results from fire needles were not different from conventional acupuncture. This result was consistent with the traditional understanding that a long duration of stroke did not present the best chances for recovery. However, only two studies were included in the subgroup analysis for which the stroke duration that was longer than six months and acupuncture depths that were more than 15mm. Thus, while the clinical implications should not be ignored, the subgroup analysis results should be treated with caution.

Based on traditional Chinese medicine theory, both conventional acupuncture and fire acupuncture act through adjusting the balance of human qi and blood, which, as a whole, produces therapeutic effects that alleviate and cure the diseases. Research has indicated that fire needle stimulation at lesions or acupoints can improve local blood circulation, enhance local tissue metabolism, and even eliminate the pathological changes in local tissues, including edema, hyperemia, exudation, adhesion, calcification, contracture, and ischemia. The results of fMRI studies have indicated that acupuncture therapy in patients with chronic hemiparetic stroke may exhibit modest improvements in upper limb function (specifically, spasticity and range of motion) by increasing ipsilesional motor cortex activity [49, 50]. Previous studies showed that fire needle acupuncture significantly increased BDNF expression, promoted endogenous NSC

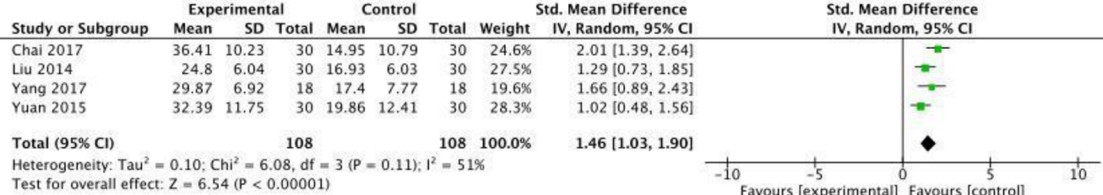

**Fig 11. Meta-analysis of fire acupuncture versus acupunture for spasticity after stroke in BI.**

**Fig 12. Meta-analysis of fire acupuncture versus acupuncture for spasticity after stroke in NDS.**

proliferation and differentiation into neurons, inhibited neuronal apoptosis, reduced inflammation by autophagy, and promoted recovery of motor neuron function [51, 52].

According to the current experimental research results and review, the mechanisms by which fire acupuncture produces antispasmodic effects after stroke are believed to be related to spasm-related neurotransmitters and receptors, and spasticity is relieved by increasing the expression of inhibitory transmitters or decreasing the expression of excitatory neurotransmitters [53]. Acupuncture also has been shown to protect central neurons in multiple ways to achieve functional restructuring of the CNS, which is crucial for strengthening central control of lower motor neurons that regulate muscle tension and to relieve muscle spasticity [54, 55].

Precise descriptions of the protocols used in acupuncture therapy are essential to enable replication and improve the transferability of the results. Generally, an acupuncture treatment plan for spasticity management should include a series of personalized and goal-oriented therapies that are tailored to the specific needs of each patient. As much as possible, we summarized the details of the acupuncture points and intervention details included in each study to provide clinicians with more options for acupuncture. The acupoints that appeared in our investigation with high frequency included the Yang-meridian, such as LI11-Quchi (nine times), LI4-Hegu (seven times), ST36-Zusanli (five times), and others. These results further verified the importance of regulating the Yang-meridian in the treatment of this disease. The result was consistent with the results of a relevant review published in 2017 [45]. Studies have shown that acupuncture in LI4 and LI11 affected local skin temperature and blood flow, while stimulation of LI11 and ST 36 were more likely to activate areas of the brain (frontal lobe, parietal lobe, sub-lobar lobe, cerebellum, and midbrain regions), causing ReHo value changes, which might promote recovery after stroke [56–58]. We also compared the key items reported in the included studies with those recommended by the Standards for Reporting Intervention in Clinical Trials of Acupuncture (STRICTA) guideline [38]. It should help clinicians and researchers to examine the standardized and more clearly described details of the studies included in this meta-analysis (Table 3). Interestingly, according to current research, acupuncture is a universally recognized non-drug treatment that can have a beneficial role in diseases that often accompany stroke, including depression, fatigue, and cognitive decline [59–61].

**Table 7. Result of meta-regression.**

| Factor | P-value | 95% Confidence interval |
|---|---|---|
| gauge of needle | 0.168 | [-0.859424,0.1974747] |
| Depth of insertion | 0.716 | [-2.718377,0.3675489] |
| Treatment Frequency | 0.594 | [-0.6249931,0.3985297] |
| Number of needle | 0.417 | [-0.5323939,1.091061] |
| Type of stroke | 0.206 | [-1.0366451,0.2878423] |
| Location of spasticity | 0.784 | [-0.5455986,0.6842484] |
| Pooled -result | 0.224 | [-1.280113, 4.283351] |

These observations also suggest that the efficacy of fire needles in the treatment of post-stroke spasticity may be underestimated.

This is the first meta-analysis to focus on the treatment of post-stroke spasms using fire needle acupuncture compared with conventional acupuncture. Although needle penetration depth and needle thickness are still controversial, our study should help to standardize fire needle treatment strategies for post-stroke spasms. We also expect our results to be adopted by policymakers and promote fire needle acupuncture as an alternative therapy to further reduce the burden of stroke on public health.

## Limitation

Several limitations of our study should be noted. First, because of the specifics of the fire needle procedure, it was not possible to carry out blind studies. Thus, the quality of the included trials was not very high. Second, the sample size was not large enough, and the number of events was small (several subgroup analyses included only two studies), which may have influenced the reliability of the conclusions and their interpretation. Third, many factors led to the moderate heterogeneity of particular outcomes in the meta-analysis process, including individual differences, varied treatment protocols (including timing, type, duration, acupoints that were used, and intensity), the stroke type, lesion location, stroke duration, and the spasticity severity. Finally, as shown by the publication dates for the literature included in this study, fire needles, which were primarily used in China, had been widely used only for the past five years in clinical practice to treat post-stroke spasticity. There are very few scholars outside of China who have focused on fire needle treatment for post-stroke spasms, resulting in the majority of participants being Chinese. This result might limit the extrapolation of our conclusions to different populations to some extent. Moreover, dissemination of the results in English would be beneficial in moving this field forward because it would help increase the interests of clinicians and researchers in using and examining the effectiveness of fire acupuncture therapies for spasticity after stroke.

## Conclusions

Although the sample size and some methodological qualities of the 16 RCTs included in the present study were not entirely satisfactory, we were able to demonstrate, to a limited extent, the efficacy of fire needle therapy for post-stroke spasticity. Acupuncture has been recommended by the World Health Organization (WHO) as an alternative and complementary strategy for stroke treatment and for improving stroke care. It is anticipated that future higher-quality RCTs will help determine the efficacy and provide reliable support for increased use of fire needles in the treatment of post-stroke spasms.

## Supporting information

**S1 Checklist. PRISMA.**
(DOCX)

**S1 Appendix. Search strategy.**
(DOCX)

## Acknowledgments

We thank Anrong Wang and sijia Cheng for helping to review the full-text articles for eligibility.

The authors would like to express their gratitude to EditSprings (https://www.editsprings.com/) for the expert linguistic services provided.

## Author Contributions

**Conceptualization:** Xuan Qiu.

**Data curation:** Xuan Qiu, Shuangmei Zhang.

**Formal analysis:** Xuan Qiu.

**Funding acquisition:** Zhaoxu Zhang.

**Investigation:** Xuan Qiu, Yicheng Gao, Shuangmei Zhang.

**Methodology:** Xuan Qiu, Yicheng Gao, Shuangmei Zhang.

**Project administration:** Zhaoxu Zhang.

**Software:** Xuan Qiu.

**Supervision:** Sijia Cheng, Shuangmei Zhang.

**Validation:** Zhaoxu Zhang, Sijia Cheng.

**Visualization:** Xuan Qiu.

**Writing – original draft:** Xuan Qiu.

**Writing – review & editing:** Xuan Qiu.

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
