## [Decision Letter · Decision Letter 0]

22 Jan 2021

PONE-D-20-29966

Fire acupuncture versus conventional acupuncture to treat spasticity after stroke

: A Systematic Review and Meta-analysis

PLOS ONE

Dear Dr. Xuan Qiu,

Thank you for submitting your manuscript to PLOS ONE. After careful consideration, we feel that it has merit but does not fully meet PLOS ONE’s publication criteria as it currently stands. Therefore, we invite you to submit a revised version of the manuscript that addresses the points raised during the review process.

The reviewers have raised a number of points which we believe major modifications are necessary to improve the manuscript, taking into account the reviewers' remarks. Please consider and address each of the comments raised by the reviewers before resubmitting the manuscript. This letter should not be construed as implying acceptance, as a revised version will be subject to re-review.

We look forward to receiving your revised manuscript.

Kind regards,

Wisit Cheungpasitporn, MD

Academic Editor

PLOS ONE

Journal Requirements:

2. Please include the date(s) on which you accessed the databases or records to obtain the data used in your study.

3. In your PRISMA checklist, please provide the page numbers where the indicated information can be found.

5. Please include your tables as part of your main manuscript and remove the individual files. Please note that supplementary tables should be uploaded as separate "supporting information" files.

Reviewers' comments:

Reviewer's Responses to Questions

**Comments to the Author**

1. Is the manuscript technically sound, and do the data support the conclusions?

Reviewer #1: Yes

Reviewer #2: Partly

2. Has the statistical analysis been performed appropriately and rigorously? 

Reviewer #1: Yes

Reviewer #2: No

3. Have the authors made all data underlying the findings in their manuscript fully available?

Reviewer #1: Yes

Reviewer #2: No

4. Is the manuscript presented in an intelligible fashion and written in standard English?

Reviewer #1: Yes

Reviewer #2: No

5. Review Comments to the Author

Reviewer #1: PLoS One

December 6th, 2020

Manuscript ID: PONE-D-20-29966

Title: Fire acupuncture versus conventional acupuncture to treat spasticity after stroke: A Systematic Review and Meta-analysis

General comments:

Thank you for the opportunity to review this timely systematic review and meta-analysis on an important and understudied topic on spasticity after stroke. The authors performed a systematic review and meta-analysis following PRISMA checklist aimed to evaluate the clinical efficacy of fire acupuncture compared with conventional acupuncture to treat post-stroke spasms and provide a detailed summary of the commonly used acupoints.

The study is well-written and well-reported according to PRISMA checklist. Also, this study brings some interesting results and new insights as a potential contribution to the field of the Complementary Therapies. I believe that this is a novel paper with a topic that will be great interest for PLoS One readers.

I have some comments, suggestions in order to strengthen the potential contribution of this topic in any revision the author(s) might undertake.

Major Revision:

METHODS

Page 6. First paragraph: after “PRISMA statement” the reference 12” must be enclosed in square brackets [12]

I would like to know how the authors get the guiding question of this systematic review? Was an acronym used, for example PICO, or PICOS, or PICOT? If yes, please provide the structure and description and reference.

Page 6. Please to use MEDLINE/PubMed instead of PubMed as database.

Page 6. Authors need to show more clearly the final key of the search strategy in each of the 8 databases accessed, mainly respecting the combinations of controlled descriptors (eg: MeSh Terms (MEDLINE / PubMed), emtree terms (EMBASE) ; etc ...) and intersections with keywords and synonyms using the Boolean AND & OR operators. I recommend that the authors present the search strategy in each database in the form of a table, containing the date on which the search was processed.

Page 7. Data collection, extraction, and management.

Make it clear in the text when data were collected in the databases.

Another important point is to report how data was extracted (what information was extracted?)

I have checked the protocol registered in the PROSPERO CRD42020188959 and it is well described there. I suggest keeping the text in that part of the article: “A piloted data extraction form that has been discussed and developed by all the reviewers will be assessed and extracted independently by two authors (ZSM and QX). A standardized form will be used to extract data, including general information, study characteristic, participant characteristic, interventions characteristics , outcomes and so on. Any disagreement in data extraction will be resolved by discussion or negotiation with a third arbitrator (ZWF). Contact the author for more information if necessary. Each eligible trial will be assigned to a study ID in the following formats: the name of the first author + space + year of publication (e.g, Wang T 2019).”

Page 7. Risk of bias assessment.

Please replace the Cochrane Bias Risk Assessment (RoB-1) to the recently updated and updated Cochrane Tool (RoB-2), since RoB-1 is coming into disuse.The internal validity and risk of bias of trials should be assesd using RoB 2 - a revised Cochrane tool assessing risk of bias arising from five domains in randomised trials: the randomisation process, deviations from the intended interventions, missing outcome data, measurement of the outcome, and selection of the reported result. Each domain a risk of bias (low risk, some concerns, or high risk) based on the domain algorithm, and made an overall judgment (low risk, some concerns or high risk) using the described criteria (Sterne et al., 2019).According to RoB 2, risk-of-bias judgments for each domain have the following categories: low risk of bias, some concerns, or high risk of bias. Judgments are based on and summarise the answers to signalling questions. RoB 2 also includes algorithms that map responses to signalling questions to a proposed risk-of-bias judgment for each domain. Response options for an overall judgment are the same as those for individual domains. The study can be judged to have (1) a low risk of bias for all domains for this result (low risk of bias), (2) raise some concerns in at least one domain for this result but not to be at high risk of bias for any domain (some concerns), or (3) have a high risk of bias in at least one domain for this result or have some concerns for multiple domains in a manner that substantially reduces confidence in the result (high risk of bias). Overall risk of bias also generally corresponds to the worst risk of bias in any of the domains. However, if a study is judged to have some concerns about risk of bias for multiple domains, it might be judged as having a high risk of bias overall. (Sterne et al., 2019).

Sterne JAC, Savović J, Page MJ, et al. RoB 2: a revised tool for assessing risk of bias in randomised trials. BMJ 2019;366:l4898. doi:10.1136/bmj.l4898.

RESULTS

Table 1. Detail of studies include

Please, put the meaning of the second column (T / C) in the figure caption: (Treatment / Control group).

Still in the second column "Age range" replaced to "Age range (M ± SD)" and put the meaning in the legend - mean and standard deviation.

Check the studies Liu et al., 2018 and Chen et al., 2005, as there is no standard deviation. It is necessary to be consistent and maintain standardization.

The study by Sheng et al 2017 the control group should be average age 56.21 instead og 5 6.21 (remove the space)

The study by Wang et al., 2018 the average age of the control group is misspelled. In the results on page 9 it is stated that ”The ages of the patients ranged from 34 to 80 years. Therefore, I believe that instead of C: 99.13 ± 33.86 it must be wrong. Please check.

DISCUSSION

Page 17. Please, consider to include the reference doi:10.1177/2515690X19834169, together with the others [53,55].

“Interestingly, according to current research, acupuncture is a universally recognized non-drug treatment that can have a beneficial role in diseases that often accompany stroke, including depression, fatigue, and cognitive decline[53-55; Abrahão et al., 2019 ].”

Reviewer #2: 1. Search terms need to provided in complete. "as an example is provided in the Supplementary material" is not acceptable. Search terms in Eight databases are different (PubMed, Web of Science, the Cochrane database, EMBASE, CBM, CNKI, WanFang, and VIP) Please attach syntax used in each database as supplementary.

2. Who are “two independent investigators”?

3. It will be better to show kappa for the selection and data extraction. Please show the data of kappa of agreement during the systematic searches. How disagreements were solved during the systematic search among two independent reviewers?

4. Please make the data for this review publicly available, possibly through the Open Science Framework (osf.io). Items to include: list of excluded studies, commands for statistical analysis, spreadsheets or data used for the meta-analyses, etc. Making data publicly available will promote the reproducibility of the review and is best practices for systematic reviews and meta-analyses.

5. Figure1, suggest to use PRISMA 2009 Flow Diagram platform

6. Forrest plots and funnel plots need to be provided.

7. Random or Fixed effect was used, needs to be specified in the abstract.

8. Authors should discuss the reason of heterogeneity.

9. There is still a considerable heterogeneity as in your limitation. Meta-regression analysis is then strongly recommended.

10. -The PICOS of the meta-analysis should be clearly reported.

6. PLOS authors have the option to publish the peer review history of their article (what does this mean?). If published, this will include your full peer review and any attached files.

Reviewer #1: **Yes: **Luís Carlos Lopes-Júnior

Reviewer #2: No

---

## [Author Response · Author response to Decision Letter 0]

3 Feb 2021

Regarding the comments of reviewers, we have responded and answered one by one in the Response to Reviewers file

---

## [Decision Letter · Decision Letter 1]

2 Mar 2021

PONE-D-20-29966R1

Fire acupuncture versus conventional acupuncture to treat spasticity after stroke

: A Systematic Review and Meta-analysis

PLOS ONE

Dear Dr. Xuan Qiu,

Thank you for submitting your manuscript to PLOS ONE. After careful consideration, we feel that it has merit but does not fully meet PLOS ONE’s publication criteria as it currently stands. Therefore, we invite you to submit a revised version of the manuscript that addresses the points raised during the review process.

ACADEMIC EDITOR: Our expert reviewer(s) have recommended some minor revisions to your manuscript. Therefore, I invite you to respond to the reviewer(s)' comments as below and revise your manuscript.

We look forward to receiving your revised manuscript.

Kind regards,

Wisit Cheungpasitporn, MD

Academic Editor

PLOS ONE

Reviewers' comments:

Reviewer's Responses to Questions

**Comments to the Author**

1. If the authors have adequately addressed your comments raised in a previous round of review and you feel that this manuscript is now acceptable for publication, you may indicate that here to bypass the “Comments to the Author” section, enter your conflict of interest statement in the “Confidential to Editor” section, and submit your "Accept" recommendation.

Reviewer #1: All comments have been addressed

Reviewer #2: All comments have been addressed

2. Is the manuscript technically sound, and do the data support the conclusions?

Reviewer #1: Yes

Reviewer #2: Yes

3. Has the statistical analysis been performed appropriately and rigorously? 

Reviewer #1: Yes

Reviewer #2: Yes

4. Have the authors made all data underlying the findings in their manuscript fully available?

Reviewer #1: Yes

Reviewer #2: Yes

5. Is the manuscript presented in an intelligible fashion and written in standard English?

Reviewer #1: No

Reviewer #2: Yes

6. Review Comments to the Author

Reviewer #1: PLoS One

February 13th, 2021

Manuscript ID: PONE-D-20-29966

Title: Fire acupuncture versus conventional acupuncture to treat spasticity after stroke: A Systematic Review and Meta-analysis

General comments:

Thank you for the opportunity to review again this timely systematic review and meta-analysis on this relevant topic

The authors have responded appropriately to all my suggestions and recommendations. The article is better presented in this version.

Just a few minor adjustments need to be made:

Minor Revision:

Abstract: Add in the “method” that the methodological evaluation or critical appraisal of the included articles was assessed using RoB-2. And in the results, to add that “according to the criteria of the RoB 2.0 tool, most of the studies are considered to have some concerns.

METHODS

Data collection, extraction, and management: write the phrase in the past instead of using the verb in the future.

Data syntheses: Same. Please use the past tense “Random effect model were used….”

RESULTS

Table 5. Please to replace the symbols “+/-” by the official of RoB-2

Low (+)

High (-)

Some concerns (?)

Also the column “Overal risk” shoud be presenting as the last column in this Table.

Ad hoc consultant.

Reviewer #2: Authors welcomed all suggestions and observations comprised in the first revision of the paper.

Authors have satisfied the comments of the reviewers

7. PLOS authors have the option to publish the peer review history of their article (what does this mean?). If published, this will include your full peer review and any attached files.

Reviewer #1: **Yes: **Prof. Dr. Luís Carlos Lopes Júnior

Reviewer #2: No

---

## [Author Response · Author response to Decision Letter 1]

4 Mar 2021

The review comments have been answered step by step, please refer to the relevant attachments for specific details

---

## [Decision Letter · Decision Letter 2]

16 Mar 2021

Fire acupuncture versus conventional acupuncture to treat spasticity after stroke

: A Systematic Review and Meta-analysis

PONE-D-20-29966R2

Dear Dr. qiu,

We’re pleased to inform you that your manuscript has been judged scientifically suitable for publication and will be formally accepted for publication once it meets all outstanding technical requirements.

Kind regards,

Wisit Cheungpasitporn, MD

Academic Editor

PLOS ONE

Additional Editor Comments:

I reviewed the revised manuscript and the response to reviewers' comments. Revised Manuscript is well written. All comments have been addressed and thus accepted for publication.

Reviewers' comments:

Reviewer's Responses to Questions

**Comments to the Author**

1. If the authors have adequately addressed your comments raised in a previous round of review and you feel that this manuscript is now acceptable for publication, you may indicate that here to bypass the “Comments to the Author” section, enter your conflict of interest statement in the “Confidential to Editor” section, and submit your "Accept" recommendation.

Reviewer #1: All comments have been addressed

Reviewer #2: All comments have been addressed

2. Is the manuscript technically sound, and do the data support the conclusions?

Reviewer #1: Yes

Reviewer #2: Yes

3. Has the statistical analysis been performed appropriately and rigorously? 

Reviewer #1: Yes

Reviewer #2: Yes

4. Have the authors made all data underlying the findings in their manuscript fully available?

Reviewer #1: Yes

Reviewer #2: Yes

5. Is the manuscript presented in an intelligible fashion and written in standard English?

Reviewer #1: Yes

Reviewer #2: Yes

6. Review Comments to the Author

Reviewer #1: The authors have responded appropriately to all my questions and recommendations.

The manuscript is better presentable right now and, therefore, I approve this version for publication at PLOS ONE.

Dr. Luís Carlos Lopes Júnior

Reviewer #2: Authors welcomed all suggestions and observations comprised in the first revision of the paper.

No further change is necessary in the opinion of the reviewer.

Authors have satisfied the comments of the reviewers

7. PLOS authors have the option to publish the peer review history of their article (what does this mean?). If published, this will include your full peer review and any attached files.

Reviewer #1: **Yes: **Prof. Dr. Luís Carlos Lopes-Júnior

Reviewer #2: No

---

## [Editor Report · Acceptance letter]

22 Mar 2021

PONE-D-20-29966R2 

Fire Acupuncture versus Conventional Acupuncture to Treat Spasticity after Stroke: A Systematic Review and Meta-analysis 

Dear Dr. Qiu:

I'm pleased to inform you that your manuscript has been deemed suitable for publication in PLOS ONE. Congratulations! Your manuscript is now with our production department. 

Kind regards, 

on behalf of

Dr. Wisit Cheungpasitporn 

Academic Editor

PLOS ONE